# CXCR1: A Cancer Stem Cell Marker and Therapeutic Target in Solid Tumors

**DOI:** 10.3390/biomedicines11020576

**Published:** 2023-02-16

**Authors:** Caitlin Molczyk, Rakesh K. Singh

**Affiliations:** Department of Pathology and Microbiology, University of Nebraska Medical Center, Omaha, NE 68198-5900, USA

**Keywords:** chemokines, cancer stem cell, CXC receptors

## Abstract

Therapy resistance is a significant concern while treating malignant disease. Accumulating evidence suggests that a subset of cancer cells potentiates tumor survival, therapy resistance, and relapse. Several different pathways regulate these purported cancer stem cells (CSCs). Evidence shows that the inflammatory tumor microenvironment plays a crucial role in maintaining the cancer stem cell pool. Typically, in the case of the tumor microenvironment, inflammatory pathways can be utilized by the tumor to aid in tumor progression; one such pathway is the CXCR1/2 pathway. The CXCR1 and CXCR2 receptors are intricately related, with CXCR1 binding two ligands that also bind CXCR2. They have the same downstream pathways but potentially separate roles in the tumor microenvironment. CXCR1 is becoming more well known for its role as a cancer stem cell identifier and therapeutic target. This review elucidates the role of the CXCR1 axis as a CSC marker in several solid tumors and discusses the utility of CXCR1 as a therapeutic target.

## 1. CXCR1 Axis in Inflammatory Tumor Microenvironment

Inflammation is essential in the body because it recruits cells to fight potential infections and heal damaged areas. There are four cardinal signs of inflammation: rubor (redness), calor (heat), tumor (swelling), and dolor (pain) [1]. The causes of these cardinal signs are microscopic, involving many cells and signaling pathways, including cytokines such as IL-1β, IL-6, TNF-α, and chemokines, with downstream signaling of NF-kB, MAPK, and JAK-STAT pathways [2]. These pathways generally lead to one of two outcomes: clearance or chronic inflammation.

Chemokines play a prominent role in inflammation. The name chemokine is short for chemotactic cytokine. Oppenheim et al. first discovered that this unique cytokine group could induce the chemotaxis of leukocytes. The first chemokine discovered is interleukin-8 (IL-8), also known as CXCL8 [3,4]. This discovery led researchers to believe that the essential role of these chemokine gradients is to influence and direct the leukocyte population [4]. Although they are primarily associated with inflammation and leukocyte chemotaxis, chemokines have a significant role in developing and maintaining the tumor microenvironment. Tumor cells can utilize the chemokines to communicate with each other and other cells in the microenvironment to benefit the tumor (Figure 1) [5,6,7,8,9].

## 2. Chemokine Superfamily

Since the discovery of CXCL8, over 40 more chemokines have been discovered. Chemokines are known for their exceptionally small molecular weight of 8–15 kD [3,10]. Four chemokine families are separated based on the cysteine residues in the N-terminus of the ligand. The ligands bind G-coupled protein receptors in the C, CC, CXC, or CX_3_C family [11,12]. This review will focus on the CXCR1 axis, one of the seven receptors in the CXC receptor family. CXCR1 has 76% homology to its sibling receptor, CXCR2. CXCR2 is the most promiscuous in the family, binding seven ligands CXCL1-3 and 5-8, whereas CXCR1 binds only CXCL6 and CXCL8 [13]. CXCR1 and CXCR2 appear to have the same downstream effects (Figure 2) that are pro-tumorigenic [14]. The known differences in these two receptors include binding affinity differences for CXCL8 and variations in the downstream effects, which have exclusively been evaluated using neutrophils [15,16].

## 3. CXCR1-Axis Chemokines

The two chemokines which bind CXCR1 are CXCL6 and CXCL8. These ligands lead to tumor cells’ pro-tumorigenic capacities, outlined in Figure 3 [14,17].

### 3.1. CXCL6

CXCL6 is also referred to as granulocyte chemotactic protein-2 (GCP-2) [18]. Interestingly, all the chemokine ligand genes are on chromosome 4. This chemokine promotes angiogenesis as an ELR^+^ chemokine with a Glu-Leu-Arg motif at the N-terminus [10]. It binds CXCR1 and CXCR2, leading to downstream effects of NF-kB, JAK-STAT, mTOR, RAS-RAF-ERK, and calcium release leading to chemotaxis [14]. The latter of these has been shown in neutrophils. The cells which express CXCL6 include tumor cells [19], fibroblasts [20], endothelial cells [17], macrophages, T cells [21], and neutrophils [22].

### 3.2. CXCL8

CXCL8 is also known as interleukin-8 (IL-8). Its gene is located on chromosome 4 and is an ELR^+^ chemokine [10]. It binds the same receptors as CXCL6 and leads to the same downstream effects [14]. This ligand is the most well-studied chemokine. Tumor cells, eosinophils, epithelial cells, fibroblasts, endothelial cells, T cells, macrophages, and neutrophils express CXCL8 [23]. The literature associates CXCL8 with tumor growth, chemoresistance, and epithelial-to-mesenchymal transition (EMT) [24].

## 4. CXCR1 and CXCR2: Potential Different Roles

The CXCR1 and CXCR2 receptors are inflammatory signaling seven-transmembrane G-coupled protein receptors that recruit and activate leukocytes. It was not until recently that these pathways were recognized as being used by the tumor cells in autocrine and paracrine manners. CXCR1 and CXCR2 are typically seen as single units in their signaling capabilities, with a 76% sequence homology and overlap of binding ligands. CXCR1 solely binds CXCL6 and CXCL8, whereas CXCR2 binds CXCL1-3 and 5-8. Both receptors are known to cause calcium release, activation of Ras/MAPK and PI3K signaling, chemotaxis, and receptor internalization [25,26].

Although CXCR1 and CXCR2 share many similarities, there are some known differences in their downstream effects, primarily evaluated in neutrophils. CXCR1 is known to induce ROS generation and oxidative bursts. This receptor also recycles to the cell surface faster than CXCR2. The internalization of CXCR2 is faster than CXCR1, even at lower ligand concentrations, when evaluated in neutrophils with the CXCL8 ligand present [27,28]. These examples are only a few characteristics separating CXCR1 from CXCR2. For other distinguishing features, please read the review by Ha et al. (2017) [28]. These varying aspects between CXCR1 and CXCR2 lead to more questions about how they function in the tumor microenvironment, specifically pertaining to how they affect tumor cells. Although small, these differences could greatly benefit the tumor cells by overexpressing one receptor over the other. For a more detailed look, specifically with CXCL8 with CXCR1 and CXCR2 pathways, the review by Mishra et al. [29] elaborates more on the signaling pathways in breast cancer.

Overall, there is yet to be an analysis of the downstream effects that differ between CXCR1 and CXCR2. There could be a difference in downstream effectors separating CXCR1 and CXCR2, which leads CXCR1 to be a rising cancer stem cell (CSC)-like marker in several solid tumors. Though there are limited reports of CXCR2 associating with CSC-like qualities in solid tumors [30,31,32].

Understanding the function of CXCR1 in promoting CSCs is essential, but this interaction is not solely between CSCs. CSCs reside in the complex TME and constantly communicate with all the inhabiting cells. The heterogeneity within the TME involving the stromal, tumor, and cancer stem cell populations affects the interplay and expression of CSC-like markers [33]. This review article summarizes our current understanding of CXCR1 and its involvement as a CSC-like marker in solid tumors. We will discuss these areas further to elucidate this receptor and its relationship with CSCs.

## 5. Cancer Stem Cells

In cancer, some reasons for high mortality include its ability to resist chemotherapy and metastasize. These highly aggressive characteristics can be attributed to a subpopulation of cells known as CSCs. CSCs are heterogeneous tumor cell populations capable of self-renewal and are considered initiators of the tumor [34], which typically constitute less than 1% of the tumor cell population. These cells are separated from the bulk tumor because they have characteristics of stem cells, such as slow cell cycling, self-renewal capability, and clonal repopulation ability [34,35,36]. They also display higher resistance to chemotherapy, allowing them to overcome chemotherapy and radiation treatment, thereby maintaining tumor growth [37].

CSCs repopulate the tumor, resist chemotherapy, and self-renew [38], suggesting they are responsible for tumor burden. Therefore, understanding the mechanisms of CSCs is essential for decreasing tumor burden and improving treatment outcomes. The most straightforward approach to decreasing CSC activity is identifying a CSC marker that can double as a therapeutic target. Currently, most known CSC markers do not act as therapeutic targets, including CD24, CD44, ALDH1, CD133, SOX2, and NANOG [39,40], though some are gaining interest as targets. Specifically, the use of drug-containing cubosomes engineered to target CD44^+^ tumor cells did succeed therapeutically in several cancer cell lines in vitro [41], with continuing research to evaluate the use of CD44 as a therapeutic target [42]. Thus far, locating a targetable CSC receptor pathway essential for their CSC characteristics has been intangible.

The complex nature of tumor cells increases the difficulty in defining CSCs. Mounting evidence suggests tumor cells are not solely in one state due to their plastic nature. In addition to plasticity, CSCs undergo asymmetrical division [43]. This type of cell division can lead to more CSCs and bulk tumor cells. We have also learned that the CSC state is most likely a phenotypic state, which means a variety of factors in the specific microenvironment can affect the transition of this state [44,45,46]. The heterogeneity and plastic nature of these cells suggest that these tumor cells may not always express the same markers [33]. This is why we typically use several markers to identify CSCs.

## 6. CSC Markers

Much of the previous research investigating CSCs relies on cell markers to locate them. Several known markers for many cancers’ CSCs include CXCR4, CD133, CD24, CD44, and the internal SOX2 and NANOG proteins used to evaluate stemness [37,47,48]. CSC markers vary from tumor to tumor, and some are highly specific to the cancer type, such as carcinoembryonic antigen (CEA) is a biomarker and potential CSC marker in colon cancer [49,50]. However, most of these markers solely identify CSCs and are not used as therapeutic targets. A promising marker of interest is CXCR1. This receptor can be both a marker and a therapeutic target. This C-X-C receptor is most known for its role in inflammatory responses. Binding CXCL6 or CXCL8 facilitates chemotaxis, promotes angiogenesis, and enhances bacterial immune system response under normal conditions [51]. In pathologic tumor conditions, the CXCR1 axis elicits a response that promotes metastasis/migration [52,53,54,55,56], invasion [26,57,58], neovascularization [8,52,56,58], increased proliferation [52,56,59], and chemotherapy resistance [53,54,55,60]. We discuss the current literature of CXCR1 as a CSC-like marker below in different malignancies.

## 7. CXCR1 Expression by CSCs

### 7.1. Breast Cancer

Breast cancer is, and continues to be, the leading cause of cancer-related deaths for women [61]. The most aggressive breast cancers remain HER^+^ and triple-negative breast cancer (TNBC). These two subtypes are infamously known for their aggressive behavior and formation of metastases, including elusive brain metastases. Breast cancer is the first tumor where CXCR1 was recognized as a CSC marker. Through patient sample analysis, the Yao research group found a correlation between breast cancer relapse, metastases, and CXCR1 expression [62]. This called into question the role of CXCR1, not its fraternal twin CXCR2, in regulating CSCs. Though there is a report of the CXCL1–CXCR2 axis being involved in TNBC CSC-like phenotypes [31], there is conflicting evidence over the importance of CXCR1 and CXCR2 in the association with CSC-like phenotype in breast cancer.

Jia et al. described a mechanism where breast cancer cells have an autocrine loop post-chemotherapy withdrawal that facilitates the regrowth of these cells. They showed the role of chemokines and cytokines, including CXCL8, with known downstream effectors NF-κB and Wnt/β-catenin. Their group found a significant decrease in CSC growth in vitro via modulation of the NF-κB and Wnt/β-catenin pathways and inhibiting the CXCR1/2 receptors with repertaxin [63].

It was relatively recent (in the early 2000s) that researchers noticed the increased expression of CXCL8 in the blood of patients with breast cancer. Interestingly, breast tumor tissue sections had elevated CXCL8 expression [64]. Later, in the 2010s, the examination of CXCL8 and CXCR1 began, leading to the evaluation of how the CXCL8–CXCR1 axis could precipitate CSC-like qualities. Ginestier et al. found that CXCR1 co-localizes with ALDEFLUOR expression in TNBC. When targeting CXCR1 with an anti-CXCR1 antibody, it decreased ALDEFLUOR-expressing cells and reduced cell viability. In their evaluation, CXCR2^+^/ALDELUOR^+^ cells made up 3% of the population. After treatment with an anti-CXCR2 antibody, there was no change in the cell viability or ALDEFLUOR^+^ cells [53]. Other labs have also found an association between CXCR1 and CD24−/CD44+ cells, which displayed CSC-like properties of colony formation [65].

The group led by Dr. Wicha continued its evaluation of the CXCR1 axis by evaluating inhibitors for this pathway, specifically repertaxin (also referred to as reperixin). By treating TNBC CSCs in vitro with the CXCR1 antagonist repertaxin, CSCs had decreased ALDH1 expression, decreased cell viability, and increased apoptosis. This group also indicated that repertaxin-treated CSCs have less potential for tumorsphere formation in vitro. When examining mice injected with TNBC cell lines, they found that repertaxin- and docetaxel-treated mice had decreased tumor burden and a decreased ALDH^+^ CSC population. In vivo xenograft experiments revealed a significant size reduction in the docetaxel- or repertaxin-treated tumors compared with the control tumors [53].

Furthermore, repertaxin- and combination-treated tumors significantly decreased ALDEFLUOR^+^ cells. Those tumors treated with docetaxel alone had an increased number of ALDEFLUOR^+^ cells. These cells were then serial diluted and orthotopically implanted into secondary mice with no treatment. The control- and docetaxel-treated tumor cells formed tumors at all dilutions. In contrast, only the lower dilutions of cells obtained from xenografts treated with repertaxin alone or in combination could generate tumors [53].

Since the discovery of CXCR1-targeted therapy reducing CSC-like phenotypes, further exploration of the downstream pathways has been explored. Brandolini et al. suggested that CXCR1’s effect on the CSC-like qualities relies on its ability to mediate the FAK/AKT pathway in breast cancer. They investigated the FAK/AKT pathway when treated with paclitaxel and repertaxin. They found there is a dose-dependent response to repertaxin by the cells. Repertaxin decreased FAK and β-catenin compared to the control [66]. FAK decreased self-renewal via decreasing Wnt/β-catenin signaling. They also showed a decrease in cell proliferation in vivo measured via proliferating cell nuclear antigen (PCNA) staining and an increase in apoptosis [67]. However, they did not stain nor show single-cell sequencing to verify a decrease in CSCs.

The effects of this antagonist in combination with chemotherapy have been assessed with docetaxel [53] and paclitaxel [66]. In combination with either drug, there was a significant reduction in tumor cell growth compared to chemotherapy alone. The joint efforts of chemotherapy with the CXCR1 antagonist in the pro-apoptotic and anti-migratory effects appear to target the CSCs robustly.

A clinical study by Goldstein et al. focused on evaluating reparixin with breast cancer CSCs. Twenty patients with HER-2 negative operable breast cancer received an oral dose of reparixin three times daily, 21 days before surgery. Via flow cytometry, they found a greater than or equal to 20% decrease in ALDH^+^ and CD24^−^/CD44^+^ cells in 9/19 and 6/19 patients, respectively. They also found a decrease in CXCR1^+^ cells. They are unsure of the clinical relevance of these numbers [65].

In 2018, the clinical trial NCT02370238, a double-blind study of paclitaxel combined with reparixin or placebo for metastatic TNBC, was conducted; the results have not shown a clear benefit of reparixin. This could be attributed to enrollment difficulty and a small sample size [68]. For the future of CXCR1-targeted therapy in breast cancer therapy, we need to see how it affects patient outcomes.

### 7.2. Colon Cancer

In the 2010s, researchers noticed a correlation between CXC chemokines and colorectal cancer. One of the first papers examining this was by Olapido et al. [69]. They examined the expression CXCL1, CXCL8, and CXCR1/2, finding that CXC-signaling may be associated with poor prognosis.

Following this finding came the first papers suggesting the CXCL8–CXCR1 axis has a role in stem cell-like qualities in colon cancer. Carpentino et al. showed that CXCL8 influenced the proliferation of colon cancer stem cells (CCSCs) [70]. This led Fisher et al. to investigate further the autocrine signaling of CXCL8–CXCR1 in the CCSCs isolated from human colorectal cancer (CRC) tissue samples. Through this research, they found there was, in fact, a correlation between high expression of CXCL8 and ALDH1 and poor patient survival in CRC. This was not found with CXCR1 and ALDH1 expression, which they attributed to the small sample size of 22 patients. They also showed that the CCSCs had CXCL8-induced dose-dependent cell migration [71]. Two other groups have reported this same phenomenon [72,73]. Expanding on this CSC-like phenotype, Fisher et al. explored the cell cycle. When CXCL8 or CXCR1 was knocked out of the CCSCs, there was a partial blockade of G1 and G2/M, leading them to look at p21, which is commonly associated with apoptosis and senescence [71]. Two other groups have found comparable results in prostate cancer [74,75].

The next step in CRC CSC research must be using a drug that targets CXCR1. Using knockout models is helpful to set the stage, but in terms of clinical use, patients will always have an expression to some degree, so we need to know what happens with a decreased expression due to targeted therapy.

### 7.3. Gastric Cancer

Gastric cancer is the second leading cause of cancer-related deaths worldwide. In the early 2000s, researchers found that CXCL8 plays a significant role in gastric cancer pathogenesis [76]. This finding is inherent in the gastrointestinal (GI) system itself because many agents within the GI system can cause irritation and inflammation, one of cancer’s critical hallmarks. The inflammatory pathways involving CXC chemokines have been involved in the development and progression of gastric cancer. Patients without gastric cancer recurrence had lower serum levels of chemokines than patients with recurrence [77].

Since noticing the potential role of CXCL8, CXCR1, and CXCR2, more scientists have explored their relationship with gastric cancer. One group found that high expression of CXCR1 in Tumor, Nodes, Metastasis (TNM) Stage II and III gastric cancer is a poor prognostic factor [78]. Wang et al. published two papers that evaluated the role of CXCR1 in the CSC-like phenotype of gastric cancer cells in vitro and in vivo. Their exploration found that CXCR1 promotes aggressive and CSC-like characteristics, including proliferation, growth, migration, and invasion through evaluation by knocking out and overexpressing CXCR1. In vivo CXCR1 knockdown-injected cells developed smaller tumors, whereas the overexpression led to larger tumors. To explore this, they investigated a mechanism of the CXCR1 downstream pathways. They discovered that the CXCR1 knockdown (KD) led to decreased phosphorylation of AKT and ERK1/2, which are known for their role in proliferation, growth, angiogenesis, invasion, and metastasis. In addition, they found that the CXCR1 KD had increased apoptosis and decreased angiogenesis [79,80].

With these findings, they moved forward with assessing the role of repertaxin in combination with the standard chemotherapy, 5-fluorouracil, in gastric cancer. Their in vitro and in vivo testing led them to conclude that these therapies synergize together without showing a synergism graph. The in vivo data are suggestive of synergism. They found that repertaxin decreased the proliferation and colony-forming of the gastric cancer CSCs. In assessing the downstream effectors AKT and ERK, they found that the combination treatment decreased their phosphorylation [79].

There is some promising exploration of the CXCR1 axis involved in the aggressiveness of gastric cancer. CXCR1 KD and repertaxin inhibition of tumor growth in vivo is vital. The current research lacks a bridge between aggressive qualities and CSC-like phenotype via other CSC markers to verify its status as a CSC marker, so it does not make the most robust case.

### 7.4. Lung Cancer

In lung cancer development, CXCL8 has been known to be an essential growth factor. It was elevated in lung tumor cells compared to normal lung tissue [81] and in chemotherapy-resistant tumor cells in vitro [82]. In adenocarcinomas, metastasis and early recurrence are linked with higher CXCR1 expression [83]. Zacharias et al. supported this claim in non-small cell lung cancer (NSCLC) patient samples, where they found CXCR1^+^ staining in all samples, while CXCR2 was only expressed in a few tumor samples [84]. By utilizing a CXCL8 analog to bind CXCR1/2, Khan et al. decreased lung tumor cell proliferation, angiogenesis, and metastasis in vivo [81]. Interestingly, Zhu et al. found that the stimulation of CXCR1 by CXCL8 perpetuates the pro-tumorigenic roles of CXCL8, but that was not the case with the CXCL8–CXCR2 axis. These together suggest that the CXCL8–CXCR1 axis plays a vital role in the self-renewal pathway.

It is not only CXCL8 enabling the self-renewal and propagation of CSC-like cells in lung cancer. Li et al. explored a mechanism through which CXCL6 is involved in lung cancer progression. In their exploration of miRNAs in cancer development, they found that CXCL6 induced downregulation of the miRNA miR-515-5p. They also found that CXCL6 is a target gene of this miRNA, so there is a positive feedback loop inhibiting the expression of the miRNA. Their findings conclude that CXCL6 overexpression enhanced lung cell survival and metastasis, while overexpression of miR-515-5p led to increased apoptosis and chemosensitivity [85]. The other miRNA of interest regarding CXCL6 and lung cancer is miRNA-101-5p. miRNA-101-5p upregulation inhibited colony formation and invasion in vitro while reducing growth in vivo. By silencing CXCL6 and inhibiting miR-101-5p, the tumor cells had decreased colony formation and invasion compared to inhibiting miR-101-5p expression alone, suggesting CXCL6 is a direct target of miR-101-5p, and the CSC-like activity is reliant on CXCL6 expression [86]. From our literature review, lung cancer appears to be the only cancer with the investigation of miRNA and CXCL6 regarding CXCR1 and CSCs.

### 7.5. Melanoma

Melanoma is a particularly aggressive cutaneous cancer with nearly 1.5 million survivors. The detection methods have significantly improved, allowing most people to be diagnosed with Stage I cancer with a 5-year survival rate of almost 100%. However, for the people diagnosed with Stage IV cancer, the survival rate drops to 34% for a 3-year relative survival rate [61]. In melanoma, both the CXCL8–CXCR2 and CXCL8–CXCR1 axes have been evaluated. Only the CXCL8–CXCR1 axis is associated with tumor cell chemotaxis [87]. CXCL8 leads to vascularization, MMP-2 activation, proliferation, and metastasis [88].

From our lab, we have seen that overexpressing CXCR1 or CXCR2 leads to increased proliferation in vitro and in vivo. A MAP kinase inhibitor negated this growth in vitro, suggesting the ERK1/2 MAP kinase pathway is the responsible downstream pathway for this pro-tumorigenic phenomenon. Overexpressing either of these receptors in vivo also increased angiogenesis [58]. In another study by our lab, Singh et al. found that targeting the CXCR1/2 axis with inhibitors SCH-479833 or Navarixin (SCH-527123) inhibited proliferation, chemotaxis/metastasis, invasion, and angiogenesis [89]. Wilson et al. detailed more substantial evidence of the CXCL8–CXCR1 axis’ involvement in CSC-like qualities of melanoma. The ABCB5 drug efflux transporter, a marker for slow-cycling melanoma cells, was shown to be related to the expression of CXCR1 but not CXCR2. They established a paracrine interaction where the CXCR1^+^/ABCB5^+^ cells secrete IL-1β. IL-1β binds IL-1R on CXCR1^−^/ABCB5^−^ cells and aids in the secretion of CXCL8, acting in a paracrine manner to bind the CXCR1^+^/ABCB5^+^ cells, which perpetuates the stem cell-like maintenance, tumorigenesis, and chemoresistance [90]. The evidence in melanoma is slowly building for the CSC-like melanoma state expressing CXCR1, leaving a significant avenue for future evaluation.

### 7.6. Neurological Malignancies

Glioblastoma (GBM) is known for its highly malignant and aggressive behavior, with a median survival of less than one year. Scientists and clinicians have been searching for more targeted and effective treatment strategies. NF-kB is reported to be aberrantly expressed in this tumor type [28]. Interestingly, CXCL8 has also been shown to have high expression. One known downstream pathway of CXCR1 is NF-kB, which may play a more significant role in GBM than previously known [91]. Raychaudhuri and Vogelbaum discovered that in GBM, the tumor cells do not express CXCR2; they only express CXCR1 while also expressing CXCL8. Scientists have debated the expression of CXCR2 on GBM, as some find there is expression at the mRNA and protein level, and some do not. Overall, this group showed a causal relationship between the CXCL8–CXCR1 activation and NF-kB expression, which is known to facilitate the invasive behavior of GBM [92].

It is known that there are many downstream effects of CXCR1 (Figure 2). Zhou et al. examined the STAT3 pathway in GBM stem cells via the regulation of neurotensin activating the CXCL8–CXCR1 pathway using neutralizing antibodies and siRNAs. They assessed the role of the CXCR1 axis on the stem cell-like qualities of GBM. Neurotensin and its receptor, NTSR, can activate EGFR, which increased CXCL8 expression leading to activation of CXCR1 and STAT3 modulation of stem cell-like qualities. They evaluated CXCR2, finding little evidence of its role in maintaining stem cell-like traits [93].

The current research in GBM suggests the role of CXCR1 as a CSC-like marker and potential target, but analysis with other co-markers is lacking. After assessing the CXCR1 co-expression with CSC markers, it will be exciting to evaluate the ability of a CXCR1 therapeutic to break the blood-brain barrier and treat GBM.

### 7.7. Pancreatic Ductal Adenocarcinoma

Pancreatic ductal adenocarcinoma (PDAC) remains the fourth leading cause of cancer-related deaths in the United States. Roughly the same number of people diagnosed with PDAC also succumb to the disease [61]. This extremely aggressive cancer is challenging to diagnose and treat due to non-specific symptomology and intrinsic resistance to current therapeutic options. In evaluating the CXCR1 inflammatory axis, little is known about its role as a CSC marker. The first paper to examine its potential by Chen et al. revealed a relationship between the known PDAC CSC markers, CD44 and CD133, and CXCR1 expression in PDAC tumor sections from patients. To investigate the importance of CXCR1 in PDAC CSCs, they grew Capan-1 PDAC cells in sphere-forming media alone and combined them with an exogenous CXCL8 ligand and an anti-CXCR1 antibody. Their results indicated that combining the CXCL8 ligand with the CXCR1 antibody decreased tumorsphere growth compared to CXCL8 treated and untreated samples. They also found increased expression of CD24 and CD44 when PDAC tumor cells were treated with exogenous CXCL8 ligand. The exogenous CXCL8 also increased their migration and invasion abilities. These properties diminished with the addition of an anti-CXCR1 antibody [54], suggesting that CXCR1 was responsible for the CSC-like phenotype.

Overall, this shows a promising initial finding of CXCR1 as a CSC marker for PDAC. If we can address a manner to infiltrate the dense stroma, a CXCR1 therapeutic could potentially target these CSCs.

## 8. Conclusions and Future Directions

The current research on the CXCR1 as a CSC marker is slowly building its case. There is robust literature on breast cancer, specifically TNBC, on the ability to target CXCR1 and the CSC population that expresses it. In other solid tumors, the possibility of CXCR1 as a CSC-like marker has increasing evidence, even in the elusive GBM. Despite growing evidence, there are still many gaps to fill in evaluating CXCR1 as a CSC marker.

To streamline future experimentations, therapeutics outside of antibodies, specifically targeting CXCR1 or CXCR2, would be helpful. With receptor specificity, the data collected will show which receptor in vitro and in vivo is responsible for the CSC-like characteristics. This is lightly investigated in TNBC in vitro using CXCR1 or CXCR2 antibodies [53] but was not investigated in each cancer type nor in vivo. The other difficulty in CSC research is the variability within the tumor, cell lines, and model organisms. These affect the outcomes and analyses for CSC markers and chemokines and their role in the CSC-like phenotype. This is explored in a thought-provoking review by Lan et al. [33].

A significant area of further exploration includes examining all the CXCR1 and CXCR2 downstream pathways elicited by each ligand in the tumor cells. There are many downstream effectors of the CXCR1/2 pathway, including the NF-kB, HIF-1, Ras-Raf-ERK, MAPK, JAK2-STAT3, Wnt/β-catenin, and mTOR, along with PLC calcium mobilization, and actin and cytoskeletal effects [14]. If we could examine the downstream effect of CXCR1 compared to CXCR2 when bound to CXCL8 or CXCL6, we could evaluate which downstream pathways elicit pro-CSC-like characteristics in the tumor cells. These effector states may be transition states, similar to what the research shows for EMT states. We postulate there would be a difference in the expression of the specific downstream pathways for a CSC-like phenotype in response to CXCR1 activation compared to CXCR2 activation.

There have been superficial examinations of these pathways on the road to defining CXCR1 as a potential CSC marker. When looking at inflammation’s role in cancer and cancer stem cells, we have much to uncover. CXCR1 provides a novel avenue for CSC identification and therapeutic targeting.

## Figures and Tables

**Figure 1 biomedicines-11-00576-f001:**
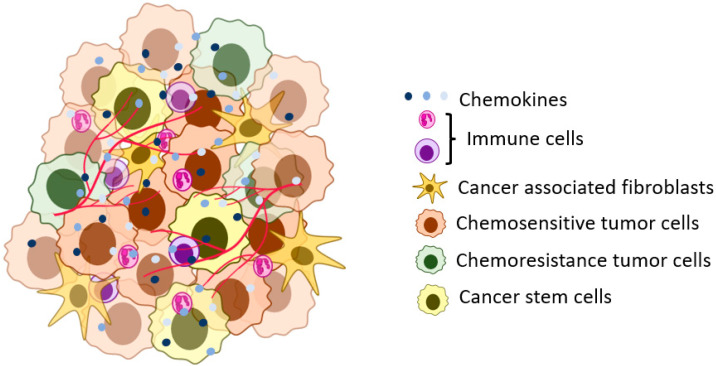
A representation of the solid tumor microenvironment. The tumor cells in the tumor microenvironment can have different phenotypic and genotypic characteristics (heterogeneity) in niche areas of the tumor. The tumor cells interact with stromal and other tumor cells using signaling pathways, including chemokine ligands.

**Figure 2 biomedicines-11-00576-f002:**
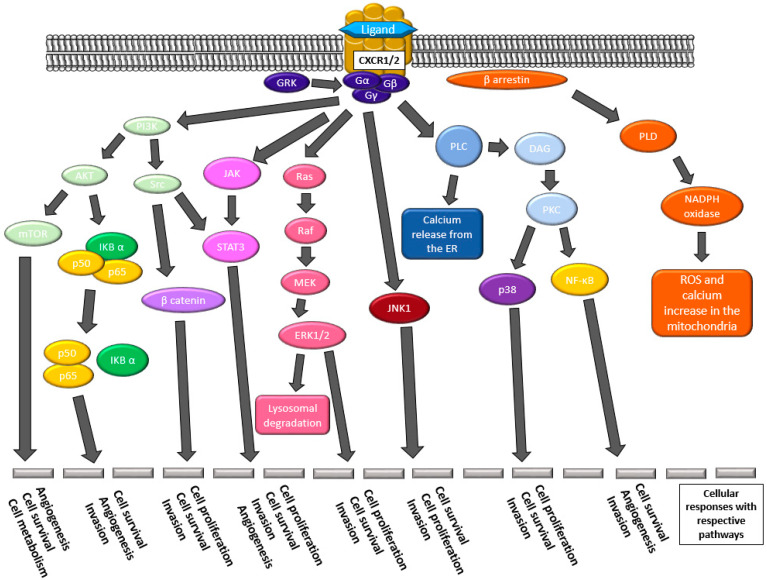
The downstream effectors of pathways elicited by the activation of CXCR1/2 by their respective ligands. Many pathways lead to angiogenesis, cellular survival and proliferation, metabolism changes, invasive properties, and increased expression of the CXCR1/2 ligands.

**Figure 3 biomedicines-11-00576-f003:**
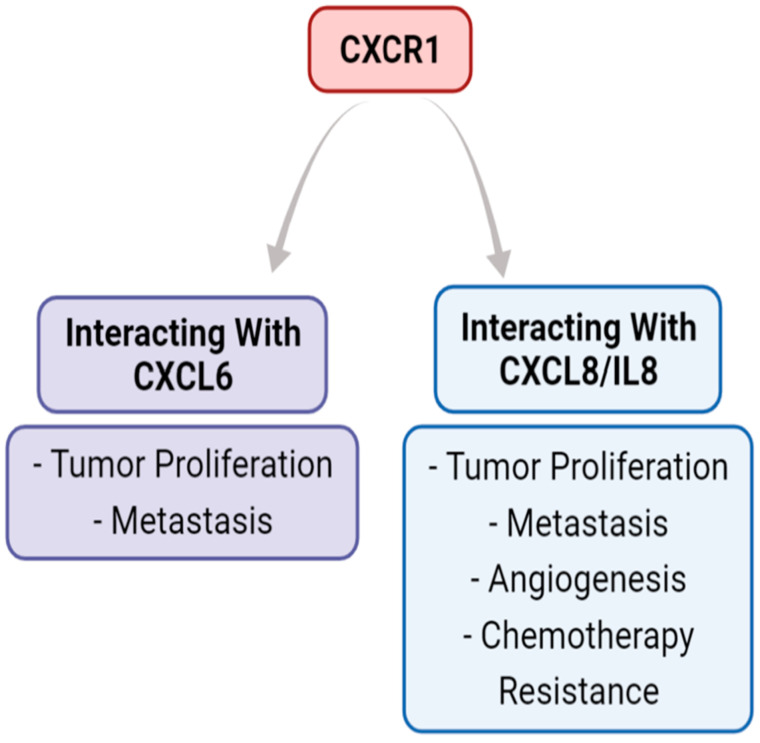
The role of the CXCR1 axis in the tumor microenvironment. The CXCR1 axis binds CXCL6 and CXCL8, but not the other five ligands that CXCR2 binds. The literature supports that CXCL6 and CXCL8 binding with CXCR1 both have pro-tumorigenic outcomes that can increase metastasis.

## Data Availability

Not applicable.

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
