# Peer review of "CXCR1: A Cancer Stem Cell Marker and Therapeutic Target in Solid Tumors"

_biomedicines, 2023, doi:10.3390/biomedicines11020576_

Round 1

Reviewer 1 Report

From the 4th paragraph (CXCR1 and CXCR2: Potential different roles), the authors claimed the CXCR1, rather than CXCR2, contributes to cancer stem cells. However, CXCR2 has been reported to promote CSCs in breast cancer (https://doi.org/10.3389/fcell.2021.689286), colon cancer ( 

https://doi.org/10.1002/jcp.27891), or hepatocellular carcinoma (https://jbiomedsci.biomedcentral.com/articles/10.1186/s12929-022-00881-4). The authors should include CXCR2 in this review article.

Author Response

We have revised our manuscript to include these studies.

Reviewer 2 Report

The authors have written a well summarized review on CXCR1/2 pathway and have illustrated that CXCR1 is gaining popularity as a target for cancer stem cell marker and as a therapeutic target in various cancers including breast cancer, lung cancer, colon cancer etc.

In line 134, the authors state that ‘most known CSC markers do not act as therapeutic targets, including CD44'.

Although CD44 receptor has been widely used as a therapeutic target for cancer stem cells and cancer cells. I would like the authors to revise the sentence and include this recent published article suggesting targeted drug delivery in triple negative breast cancer and colon cancer cell (https://doi.org/10.1021/acs.molpharmaceut.2c00439).

In section for colon cancer section 7.2, authors could also highlight the carcinoembryonic antigen (CEA) which has also been used for targeting CSCs.

Author Response

In line 134, the authors state that ‘most known CSC markers do not act as therapeutic targets, including CD44'. Although CD44 receptor has been widely used as a therapeutic target for cancer stem cells and cancer cells. I would like the authors to revise the sentence and include this recent published article suggesting targeted drug delivery in triple negative breast cancer and colon cancer cell (https://doi.org/10.1021/acs.molpharmaceut.2c00439).

We have revised our manuscript to discuss it.

In section for colon cancer section 7.2, authors could also highlight the carcinoembryonic antigen (CEA) which has also been used for targeting CSCs.

Necessary changes have been made in the revised manuscript.

Round 2

Reviewer 1 Report

The authors have greatly improved their manuscript, and I have no more questions.